# YOLOv8 Model for Weed Detection in Wheat Fields Based on a Visual Converter and Multi-Scale Feature Fusion

**DOI:** 10.3390/s24134379

**Published:** 2024-07-05

**Authors:** Yinzeng Liu, Fandi Zeng, Hongwei Diao, Junke Zhu, Dong Ji, Xijie Liao, Zhihuan Zhao

**Affiliations:** 1Mechanical and Electronic Engineering College, Shandong Agriculture and Engineering University, Jinan 250100, China; lyz19971024@163.com (Y.L.); zfd19508@163.com (F.Z.); dhw_0823@163.com (H.D.); 13325259481@163.com (D.J.); 17853435639@163.com (X.L.); 2School of Agricultural Engineering and Food Science, Shandong University of Technology, Zibo 255000, China; zhujunke@sdu.edu.cn

**Keywords:** deep learning, wheat weeds, weed detection, MobileViTv3, BiFPN

## Abstract

Accurate weed detection is essential for the precise control of weeds in wheat fields, but weeds and wheat are sheltered from each other, and there is no clear size specification, making it difficult to accurately detect weeds in wheat. To achieve the precise identification of weeds, wheat weed datasets were constructed, and a wheat field weed detection model, YOLOv8-MBM, based on improved YOLOv8s, was proposed. In this study, a lightweight visual converter (MobileViTv3) was introduced into the C2f module to enhance the detection accuracy of the model by integrating input, local (CNN), and global (ViT) features. Secondly, a bidirectional feature pyramid network (BiFPN) was introduced to enhance the performance of multi-scale feature fusion. Furthermore, to address the weak generalization and slow convergence speed of the CIoU loss function for detection tasks, the bounding box regression loss function (MPDIOU) was used instead of the CIoU loss function to improve the convergence speed of the model and further enhance the detection performance. Finally, the model performance was tested on the wheat weed datasets. The experiments show that the YOLOv8-MBM proposed in this paper is superior to Fast R-CNN, YOLOv3, YOLOv4-tiny, YOLOv5s, YOLOv7, YOLOv9, and other mainstream models in regards to detection performance. The accuracy of the improved model reaches 92.7%. Compared with the original YOLOv8s model, the precision, recall, mAP1, and mAP2 are increased by 10.6%, 8.9%, 9.7%, and 9.3%, respectively. In summary, the YOLOv8-MBM model successfully meets the requirements for accurate weed detection in wheat fields.

## 1. Introduction

Wheat is a vital staple crop that fulfills the dietary and nutritional requirements of human beings. China cultivates around 350 million mu of wheat annually, which represents 19.86% of the total area used for grain crops in the country. Wheat is the third most widely grown crop in China, after corn and rice, and it plays a crucial role in the country’s agricultural production and national economy. Nevertheless, the productivity of wheat is limited by a range of factors, with weed infestation being notably significant. Based on the survey, the predominant weeds found in wheat fields in Shandong are wheat Artemisia. These weeds directly compete with wheat for essential resources such as soil nutrients, water, and sunlight, thus significantly impeding the growth and development of wheat. Failure to manage the growth of weeds alongside wheat may result in a decrease in wheat production exceeding 34% [1,2]. Simultaneously, the presence of weeds can also amplify the prevalence of wheat pests and diseases, resulting in substantial financial losses for agricultural productivity. Early and mid-March (the green-up period) is a critical time for weeding wheat. Therefore, it is particularly important to control the growth of weeds in wheat fields during the green-up period.

The most commonly used methods of weeding in wheat fields are manual weeding and spraying herbicides [3,4]. Artificial weeding is a traditional way of weeding in wheat fields. This method exhibits high labour intensity, high weeding cost, and is susceptible to subjective influence. The efficiency of weeding is low, and it is difficult for this method to meet the needs of large-scale agricultural production. Spraying herbicides is a convenient and quick weeding method, and it can effectively control the growth of weeds in wheat fields [5,6]. However, spraying herbicides has a negative impact on the environment and crops, and can affect the health of farmers and consumers [7]. Computer vision technology is increasingly being used for crop detection and weed management. The automated weed detection technique utilizes computer vision technology to swiftly identify wheat and weeds. However, wheat and weeds are densely distributed, with similar colours and mutual occlusion. Thus, the crucial prerequisite for the successful implementation of novel weed control techniques, like precision application, mechanical weeding, and laser weeding, is the precision and accuracy of weed identification.

Both traditional machine vision and deep learning-based machine vision demonstrate high performance in regards to weed detection [8]. Conventional machine vision techniques require the extraction of feature information from the weed target, including its shape, color, texture, etc. This information is then used by a classifier to identify and detect weeds [9,10,11,12]. Tang et al. [13] used the k-means algorithm to realize the automatic identification of weeds in the field. Zhang et al. [14] proposed a weed species identification method based on GrabCut and the adaptive fuzzy dynamic k-means algorithm. Experiments show that compared with other weed identification methods, this method can automatically segment the background and avoid the problem of local gradient disappearance. Bakhshipour et al. [15] proposed a method of combining support vector machine and artificial neural network models to identify weeds in sugar beet fields. Experiments show that the overall classification accuracy of this method is 92.92%. Espejo-Garcia et al. [16] proposed a weed recognition system based on a combination of convolutional networks and classifiers. Experiments show that the F1 value of this method can reach more than 95%.

Traditional machine vision methods are complex, time-consuming, computationally inefficient, and possess high model complexity, which are suitable for specific scenarios and tasks, but show poor robustness in complex agricultural environments. Deep learning-based methods automatically extract features from images through models such as convolutional neural networks (CNNs) and complete weed detection in wheat fields through “end-to-end” training, which offers higher accuracy and robustness [17,18,19,20,21] Compared with traditional machine vision methods, deep learning-based machine vision methods are more suitable for complex and variable agricultural scenarios. Deep learning-based target detection models are divided into two categories: one-stage models and two-stage models, where the two-stage model first generates candidate regions and then performs further classification and localisation on these candidate regions. Dang et al. [22] established 18 extensive benchmarks of YOLO target detection models for weed detection. Experiments showed that the YOLO model has great potential for real-time weed identification. Wang et al. [23] proposed a seedling detection method for Solanum rostratum Dunal, based on the combination of YOLOv5 and an attention mechanism, and the real-time recognition accuracy reached 94.65%. Ajayi et al. [24] proposed a performance evaluation of automatic crop and weed classification on UAV images using the YOLOv5 model, and the combined recognition accuracy of this method for sugarcane, banana tree, pineapple, chili, and weeds can reach 78%. Zhu et al. [25] proposed a YOLOX weed recognition algorithm based on a lightweight attention module, which achieves 92.45% recognition accuracy for corn seedlings and 88.94% for weeds. Huang et al. [26] proposed a full convolutional network based on UAV RGB images to achieve semantic segmentation of weeds, and the recognition rate of weeds was 88.3%. Sudars et al. [27] compared different single-stage and two-stage models on the public dataset DeepWeeds and obtained the highest detection accuracy using Faster R-CNN, with ResNet101 as the feature extraction network. Gabriel Alberto et al. [28] used Mask-RCNN to identify weeds in sugarcane crops, and the model achieved 80.3% accuracy using ResNet—50 and ResNet—101 as backbone networks. This type of method offers high accuracy, but a long detection time when compared to one-stage models, whereas one-stage algorithms can be improved to increase the accuracy with a shorter detection time, which is more in line with the requirements for the real-time identification of weeds.

In summary, deep learning-based machine vision methods can achieve a certain degree of accuracy in weed detection. However, in the complex wheat farmland environment, problems such as similar colours of wheat crops and weeds and inconsistent size and dimensions lead to lower accuracy of the detection model and easy loss of small targets.

In order to avoid the above problems, an improved model YOLOv8-MBM (YOLOv8s-C2f_MobileViTv3-BiFPN-MPDIoU), based on YOLOv8s, is proposed to realize the target detection of wheat weeds. In this study, the C2f module in the backbone feature extraction network of YOLOv8s is improved. The lightweight visual converter (MobileViTv3) is introduced into the C2f module, and the input, local (CNN), and global (ViT) features are fused to improve the detection accuracy of the model. Secondly, a weighted bidirectional feature pyramid network (BiFPN) is introduced to enhance the performance of multi-scale feature fusion and improve the accuracy of small target detection. Finally, in order to make up for the problem of weak generalization and slow convergence speed of the CIoU loss function in the detection task, the bounding box regression loss function (MPDIoU) is used instead of the CIoU loss function to improve the convergence speed of the model, to further enhance its detection performance, and to achieve accurate detection of weed targets in complex environments.

## 2. Materials and Methods

### 2.1. Wheat Weed Dataset

The YOLOv8-MBM algorithm model training proposed in this paper must label wheat weeds, but the current open source dataset does not contain these labels. Therefore, wheat weed datasets were constructed by collecting images on site, and weeds in the dataset were labelled (with one weed in a bounding box). Due to the collection area, the only weed species in the wheat field is wheat Artemisia, so the wheat weed dataset constructed in this paper contains only one weed type, labelled as ‘wheat Artemisia’.

#### 2.1.1. Weed Image Acquisition

The wheat and weed data utilized in this study were acquired from agricultural land in Dezhou City, Shandong Province, China. To enhance the recognition model’s ability to generalize and reduce the impact of light intensity on the detection results, RGB (red, green, and blue) images under different lighting conditions were collected at 7 a.m., 12 p.m., and 6 p.m. in early and mid-March. The cameras were positioned at elevations of approximately 50 cm and 100 cm above the ground, with an inclination of 80° to 90° relative to the ground. A total of 4536 photos were captured and saved in the JPG format. The photographs were obtained using a SONY DSC-WX7 camera, with a resolution of 4608 × 3456 pixels. The captured images may be seen in Figure 1.

#### 2.1.2. Image Annotation and Dataset Construction

The Labellmg tool (https://github.com/tzutalin/labelImg, accessed on 20 January 2023.) was used to label the weed target area in the image. The labelling information primarily includes the picture file name, image dimensions, and region-specific details, such as the label, top-left pixel coordinates, and bottom-right pixel coordinates. The labelling is shown in Figure 2. To assess the effectiveness of the training model, the labelled dataset is split into three sets: the training set, the validation set, and the test set. These sets are separated in a ratio of 6:2:2.

#### 2.1.3. Dataset Expansion

In order to reduce the network overfitting and improve the generalization and robustness of the detection model, this study uses the OpenCV image processing algorithm to enhance the original image to expand the dataset. Image enhancement methods include multi-angle rotation (45°, 90°), mirror enhancement (horizontal, vertical), Gaussian noise, and salt and pepper noise techniques, as shown in Figure 3. Finally, the number of samples in the training set, validation set, and test set was expanded to 16,330, 5443, and 5443, respectively.

### 2.2. Deep Learning Network Construction

#### 2.2.1. The YOLOv8 Network Model

YOLO (You Only Look Once) is a real-time target identification algorithm that operates in a single stage. It is known for its simplicity, efficiency, and capacity for application in real-time circumstances. YOLO is extensively utilized in the field of agriculture [29,30]. YOLOv8, developed by Ultralytics, is a state-of-the-art (SOTA) model that shares the same reasoning process as that of YOLOv5. However, YOLOv8 has undergone architectural updates and enhancements. The algorithm models are classed based on their size as YOLOv8n, YOLOv8s, YOLOv8m, YOLOv8l, and YOLOv8x. Every YOLOv8 model is trained using the dataset mentioned in this research paper, and the precise parameters can be found in Table 1. The YOLOv8n model has the smallest parameter count, but it lacks sufficient residual structures, leading to lower detection accuracy. On the other hand, the YOLOv8m, YOLOv8l, and YOLOv8x models have a higher number of parameters due to an abundance of residual structures. However, this increase in parameters does not proportionally improve their mAP50 performance. This study proposes the construction of a new recognition model method based on YOLOv8s, taking into consideration the complicated farming environment, detection accuracy, and light weighting.

The YOLOv8 network consists of four parts: the Input, Backbone, Neck, and Head. Input selects the Mosaic data enhancement method, and for models of different sizes, some hyperparameters will be modified to enrich the dataset and improve the generalization ability and robustness of the model. The Backbone part uses a cross-stage local network structure to reduce the calculation amount and enhance the gradient. The spatial pyramid pooling module is used to better extract spatial features. The Neck part first performs a downsampling operation, and then an upsampling operation, so that the model offers better adaptability to targets of different sizes and shapes. The Head part adopts a decoupled head structure, which effectively reduces the number of parameters and the computational complexity and enhances the generalization ability and robustness of the model. At the same time, YOLOv8 uses the anchor-free node detection method to directly predict the centre point and width–height ratio of the target, to reduce the number of anchor frames, and to further improve the detection speed and accuracy of the model.

To enhance the accuracy of the detection model, this study incorporates a lightweight vision converter (MobileViTv3) into the C2f module, creating the C2f-MobileViTv3 module. Additionally, PANet is replaced with BiFPN to improve the accuracy of detecting small targets by enhancing multi-scale feature fusion. To address the weak generalization and slow convergence of the CIoU loss function in the detection task, the bounding box regression loss function (MPDIoU) is used instead. The final network structure is illustrated in Figure 4.

#### 2.2.2. Residual Module Combined with Vision Converter

To address the unstructured and complex environment of wheat fields and to solve the problem of the low accuracy of weed detection, this study improves these elements using the C2f module of YOLOv8s and introduces the visual converter structure to form the C2f-MobileViTv3 module to improve the detection accuracy of the model, which is shown in Figure 5, with the red box indicating the introduced visual converter structure.

Figure 6 displays the schematic diagram of the MobileViTv3 module. The module comprises a profound 3 × 3 convolutional layer (dwconv3 × 3), a 1 × 1 convolutional layer (conv1 × 1), and a linear transformer. The 1 × 1 convolutional layer is employed in the fusion block to effectively combine the local and global features without considering other locations in the feature map. This simplifies the learning process of the fusion block. By using the 1 × 1 convolutional layer, the number of parameters and FLOPs (floating point operations) remains unchanged, even when the module’s width is altered. Due to the close relationship between the features of the local representation module and the global representation module, and the slightly higher output channel of the local representation block compared to the input channel, the MobileViTv3 module combines the local representation module and the global representation module. The input features are combined with the output of the 1 × 1 convolutional layer in the fusion block. To further decrease the parameters, the local representation block employs a deep 3 × 3 convolutional layer.

#### 2.2.3. Feature Fusion Networks

The YOLOv8s network incorporates PANet as the Neck component to effectively merge targets of different scales. The structure of PANet, as depicted in Figure 7b, includes a bottom-up path, based on FPN (as shown in Figure 7a). This enables the transmission of more powerful semantic information from deeper features to the shallow feature layer, while also transmitting stronger localization information from the shallow feature layer to the deeper features. As a result, the fusion of features at different scales can be achieved. PANet is also capable of integrating many feature layers, but its implementation involves adding distinct features. However, in the case of the weed image in the wheat field, the weeds vary in size. This simple addition method results in an unequal contribution of the diverse input features to the fused output features. In order to address this issue and enhance the YOLOv8s network’s ability to capture multi-scale feature information, this research develops a weighted bidirectional feature pyramid network (BiFPN) [31]. The construction of this network is depicted in Figure 7c.

BiFPN is based on PANet, with the removal of an input node that contributes less to the overall feature network, and the addition of an extra edge between the original input and output nodes at the same level. This makes it possible to fuse more features without adding too much cost, effectively mitigating the feature loss phenomenon which can occur due to too many network levels, and finally, constructing the top-down and bottom-up fusion as a single module so that it can be stacked repeatedly to enhance the information fusion, as shown in Equation (1):(1)$$O=\sum_{i}\frac{w_{i}}{\epsilon +\sum_{j}w_{j}}\cdot I_{i}$$

In the formula, *O* represents the weight, using the activation function ReLU to ensure *w_i_* ≥ 0, *w_i_* represents the learning weight corresponding to the input feature *I_i_*, and *ε* = 0.0001. BiFPN combines two-way cross-scale connection and fast normalized fusion. Taking the feature layer P4 as an example, the two fusion processes are as follows:(2)$$P_{4}^{td}=Conv(\frac{\omega_{1}\cdot P_{4}^{in}+\omega_{2}\cdot {Resize(P}_{5}^{in})}{\omega_{1}+\omega_{2}+\epsilon })$$
(3)$$P_{4}^{out}=Conv(\frac{\omega_{1}^{\prime }\cdot P_{4}^{in}+\omega_{2}^{\prime }\cdot P_{4}^{td}+\omega_{3}^{\prime }\cdot {Resize(P}_{3}^{out})}{\omega_{1}^{\prime }+\omega_{2}^{\prime }+\omega_{3}^{\prime }+\epsilon })$$

In the formula, $P_{4}^{td}$ is the intermediate feature layer of layer 4 of the top-down path, and $P_{4}^{out}$ is the output feature layer of the bottom-up path of layer 4. In this study, BiFPN is used to replace PANet to enhance the multi-scale feature fusion performance and further improve the target detection performance.

#### 2.2.4. Bounding Box Regression Loss Function (MPDIOU)

Due to the complex environment of wheat cultivation, weeds and wheat are similar in colour and mutual occlusion, which makes it difficult to detect weeds. The CIOU loss function considers factors such as centre distance, perpendicular ratio, and overlapping area. However, during the bounding box regression process, if the predicted box has the same length and width ratio as the real box, but different width and height values, optimization issues occur, and these cannot be easily resolved. Therefore, in this paper, the bounding box regression loss function (MPDIOU) is selected to replace the original loss function, which is shown in Equation (4):(4)$$MPDIoU=IoU-\frac{\rho^{2}(P_{1}^{pred},P_{1}^{gt})}{\omega^{2}+h^{2}}-\frac{\rho^{2}(P_{2}^{pred},P_{2}^{gt})}{\omega^{2}+h^{2}}$$

$P_{1}^{pred}$, $P_{2}^{pred}$, $P_{1}^{gt}$, and $P_{2}^{gt}$ represent the points located in the upper left and lower right corners of the prediction box and the real box, respectively. The distance between the corresponding points is calculated using $\rho^{2}(P_{1}^{pred},P_{1}^{gt})$, $\rho^{2}(P_{2}^{pred},P_{2}^{gt})$.

MPDIOU simplifies the similarity comparison between two bounding boxes by minimizing the upper left and lower right point distances between the prediction box and the bounding box. The utilization of MPDIOU can successfully address the issue of detecting frame distortion resulting from the overlap of wheat and weeds, thereby significantly enhancing the weed recognition accuracy. 

### 2.3. Experimental Environment

The operating environment of the improved YOLOv8s model is shown in Table 2. In this study, the initial learning rate was set to 0.01, the final learning rate was 0.01, the weight decay coefficient was 0.0005, the intersection–union ratio (IoU) threshold was 0.7, the momentum (momentum) was set to 0.9, the learning rate adjustment strategy used was STEPS, the image size of the input network was set to 640 pixels × 640 pixels, the batch-size size was set to 32, and the number of training times was set to 300.

### 2.4. Assessment of Indicators

In this paper, we employ five metrics, namely P (precision, %), R (pecall, %), mAP1 (average mAP value on IoU thresholds of 0.5 or lower), mAP2 (average mAP value on IoU thresholds ranging from 0.5 to 0.95 at intervals of 0.05), and the value of F1 parameters, to assess the effectiveness of the network model. The calculation formula is as follows:(5)$$P=\frac{\mathrm{TP}}{\mathrm{TP}+\mathrm{FP}}\times 100\%$$
(6)$$R=\frac{\mathrm{TP}}{\mathrm{TP}+\mathrm{FN}}\times 100\%$$
(7)$$\mathrm{AP}=\int_{0}^{1}P(R)\mathrm{dR}$$
(8)$$\mathrm{mAP}=\frac{1}{M}\sum_{k=1}^{M}\mathrm{AP}\left(k\right)\times 100\%$$
(9)$$F1=\frac{P*R*2}{P+R}$$

In the formula: *TP* is the positive sample predicted as positive; *FP* is the negative sample predicted as positive; *FN* is the positive sample predicted as negative; *TN* is the negative sample predicting as negative; *k* is the current category; *M* is the number of categories.

## 3. Results and Discussion

### 3.1. Analysis of Ablation Experiments

Using the same training set, validation set, and test set, ablation experiments are conducted on different optimisation algorithms to assess and validate the effectiveness of the improved method used in this study. In this study, based on the YOLOv8s network structure, the visual converter module is introduced into the C2f module, the weighted bidirectional feature pyramid network (BiFPN) is used for multi-scale feature fusion, the CIoU loss function is replaced with the MPDIou loss function for the test, and the different optimisation algorithms are used to obtain the best model after 300 iterations to evaluate the optimisation effect of the different optimisation algorithms; the optimisation effect of the different optimisation algorithms is then evaluated. The performance index parameters of different models are shown in Table 3.

As can be seen from Table 2, the introduction of C2f-MobileViTv3, BiFPN, and MPDIoU into the original YOLOv8s network model shows different degrees of gains in regards to accuracy, recall, and average precision. The recognition accuracy of the original YOLOv8s for weeds is 82.1%, and the recognition accuracy of weeds using the C2f-MobileViTv3 module, BiFPN, and MPDIoU loss functions alone reaches 88.7%, 87.5%, and 85.6%, respectively, which is 6.6%, 5.4%, and 3.5% higher compared to that of the original model. Additionally, compared to the original model, the weight size increased by 6.5%, 5.6%, and 5.1% compared to that of the original model, indicating that the C2f-MobileViTv3 module results in a large gain in the detection accuracy of the model, and there is only a small increase in the size of the model weights, but the use of an optimisation algorithm alone is limited in improving the performance of the network model.

There is a significant improvement in the accuracy of the detection model when both optimisation methods are used, as can be seen in Table 3, where the model accuracy is increased by 10.5% when the C2f-MobileViTv3 module is used together with the BiFPN structure and the model weight size is increased by 7.5%, and the accuracy is increased by 9% when the C2f-MobileViTv3 module is co-optimised with the MPDIoU loss function.

The above experiments show that optimising the Backbone network, Neck network, and loss function of YOLOv8s can significantly increase the model recognition accuracy of weeds in the wheat field, and the novel algorithm of YOLOv8s+ C2f-MobileViTv3 + BiFPN + MPDIoU proposed in this study improves the recognition accuracy of weeds by 10.6%, the recall by 8.9%, the mAP1 by 9.7%, and the mAP2 by 9.3% when compared with the results of the original model. Figure 8 shows the performance index curves of different combinations of optimisation algorithms, where (a) is the comparison of the accuracy curves and (b) is the comparison of the recall curves. As can be seen from the figure showing the YOLOv8-MBM algorithm in the training process, its accuracy rate for value fluctuation in the first 30 training iterations is larger, showing a growth state. After completing 80 training iterations, the fluctuation tends to stabilise, and the change fluctuation is smaller; the recall rate for the first 50 training iterations exhibits a growth state, and after completing 50 training iterations, the fluctuation amplitude is reduced, gradually tending to stabilise. 

This study evaluates the performance variation for the models of different optimisation algorithms on the training and validation sets based on the loss curves, as shown in Figure 9, where (a) is the training set anchor frame loss curve, (b) is the training set target loss curve, (c) is the validation set anchor frame loss curve, and (d) is the validation set target loss curve. As can be seen from Figure 8, during the training process, the anchor frame loss value of the validation set and the target loss value of the validation set exhibited a decreasing stage in the first 50 iterations until the anchor frame loss value and the target loss value tended to smooth and stabilise after the completion of 50 iterations. The decrease in the anchor frame loss value and the target loss value of the training set tends to stabilise after 200 iterations, which indicates that the model has already been fitted by the completion of the 200th iteration. After fitting, the anchor frame loss and target loss of the validation set of the YOLOv8-MBM algorithm proposed in this study converged faster during training compared to other optimisation algorithms, and the loss values dropped to a lower point, indicating that the introduction of the C2f-MobileViTv3 module, the BiFPN structure, and the MPDIoU loss function in the YOLOv8s network showed a certain gain in the wheat field weed detection model’s training effect.

### 3.2. Visualisation of Target Area Model Features

Wheat and weeds are similar in colour, but there are obvious differences in leaf shape and stem structure. In order to verify whether the proposed YOLOv8 _ MBM algorithm also pays attention to the leaf and stem area when identifying weeds in wheat fields, this study uses Grad-CAM technology to visually interpret the YOLOv8 model of visual converter and multi-scale feature fusion to predict the weed target area and visually evaluate the weed recognition method proposed in this paper. The Grad-CAM technology uses the back propagation of training weights to perform the global average pooling of the obtained gradient matrix in the spatial dimension, and after the weighted activation of each channel in the feature layer, the heat map is obtained. The brightness depth of a certain area in the heat map can show the portion of the image that has a greater impact on the model output. The heat map of weed identification obtained by the YOLOv8-MBM model is shown in Figure 10. It can be seen from Figure 9 that the attention of the model proposed in this study is mainly focused on the stem and leaf areas in the image when identifying weeds, which are brighter in colour and more responsive to the imaging, while the original YOLOv8s model focuses on the leaves when identifying weeds, and thus, can miss their detection. The heat map visualization experiment regarding weed identification proves that the image information based on the YOLOv8-MBM model proposed in this study is validated for identifying weeds in wheat fields.

### 3.3. Experimental Analysis of Different Modelling Algorithms

In order to further evaluate the effectiveness of the YOLOv8s-MBM algorithm for weed target detection in wheat fields, the current mainstream Fast-RCNN and YOLO target detection algorithms are trained by deep learning using the same training set, verification set, and test set. The accuracy of the Fast-RCNN network model is 82.7%, and the frame rate is 6.5 FPS, which is quite different from the accuracy of the algorithm proposed in this paper. The YOLO series involved in the comparative detection network are: YOLOv3, YOLOv4-tiny, YOLOv5 s, YOLOv7, and YOLOv9, the test results are shown in Table 4.

As can be seen from Table 4, the YOLOv8s-MBM algorithm achieves 92.7%, 87.6%, 89.7%, 85.2%, and 35.5 FPS for P, R, mAP1, mAP2, and FPS values, respectively, and these rates show the advantage of this method over the similar YOLO model algorithms. The YOLOv8s-MBM algorithm is capable of extracting the key feature information in a complex wheat field environment, and it improves the detection accuracy of the model by introducing a visual in the C2f module converter module, fusing input, and local and global features to improve the detection accuracy of the model, replacing the Neck network with a weighted bidirectional feature pyramid network (BiFPN) to enhance the performance of multi-scale feature fusion and improve the detection accuracy of small targets. Therefore, the network is capable of accurately identifying weed targets in a complex wheat field environment.

### 3.4. Weed Detection Effect of YOLOv8-MBM Algorithm

In order to verify the effectiveness of the YOLOv8-MBM algorithm proposed in this study in recognizing weeds in wheat fields during the wheat green-up period under natural conditions, i.e., four cases of good near-distance light, bad near-distance light, good long-distance light, and bad long-distance light, were randomly selected for target detection. The detection effect is shown in Figure 11, in which it can be seen that the improved model exhibits a good recognition ability in the natural environment. Only a very small number of weeds cannot be detected due to occlusion or a particularly small target size. In most cases, the algorithm can detect weeds. In summary, the YOLOv8-MBM algorithm proposed in this study shows good recognition ability and can effectively detect weeds in wheat fields in the natural environment.

## 4. Conclusions

(1) In this paper, we address the problem of similarity in colour and inconsistency in size and dimensions of wheat crops and weeds in regards to weed identification in complex wheat field environments during the green-up period. We propose an improved model YOLOv8s-based YOLOv8-MBM, which improves the C2f module in the Backbone feature extraction network of YOLOv8s and introduces a lightweight visual converter (MobileViTv3) into the C2f module. Secondly, the weighted bidirectional feature pyramid network (BiFPN) is introduced to enhance the multi-scale feature fusion performance and improve the detection accuracy for small targets. Finally, to compensate for the weak generalisation and slow convergence of the CIoU loss function in the detection task, a bounding box regression loss function (MPDIoU) is used instead of the CIoU loss function to improve the convergence speed of the model, further enhance the model detection performance, and achieve accurate detection of weed targets in complex environments.

(2) Different optimisation algorithms are trained using the same training set and validation set, and the same test set is used to compare the four parameters of different optimisation algorithms, such as the precision, recall, mAP1 (mAP@0.5), and mAP2 (mAP@0.50-0.95), and the experimental comparisons show that the proposed algorithm achieves an accuracy rate of 92.7%. Compared with the original YOLOv8s model, the precision, recall, mAP1, and mAP2 are increased by 10.6%, 8.9%, 9.7%, and 9.3%, respectively. Meanwhile, comparing with YOLO series algorithms, the experiment shows that the accuracy of the YOLOv8-MBM algorithm is higher than that of the YOLOv3, YOLOv4-tiny, YOLOv5s, YOLOv7, and YOLOv9 algorithms. The FPS of the proposed algorithm reaches 35.5, which meets the requirements for real-time detection. In summary, the algorithm proposed in this paper is more suitable for weed detection in complex farmland environments.

(3) In future studies, we will increase the data for weeds at different growth stages, in different planting areas, and for different species. Currently, we have only studied one weed species, wheat Artemisia. At a later stage, we will carry out the identification of all weed species in wheat fields to improve the general applicability of the model, as well as to further improve the detection accuracy.

## Figures and Tables

**Figure 1 sensors-24-04379-f001:**
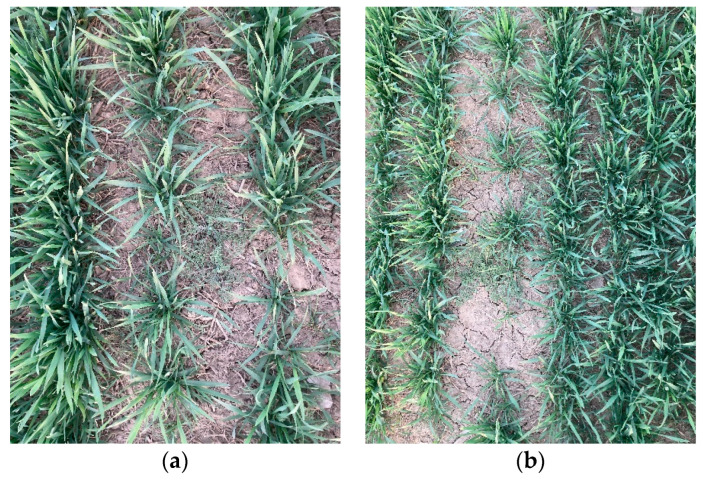
Image data captured. (**a**) near-distance (**b**) long-distance.

**Figure 2 sensors-24-04379-f002:**
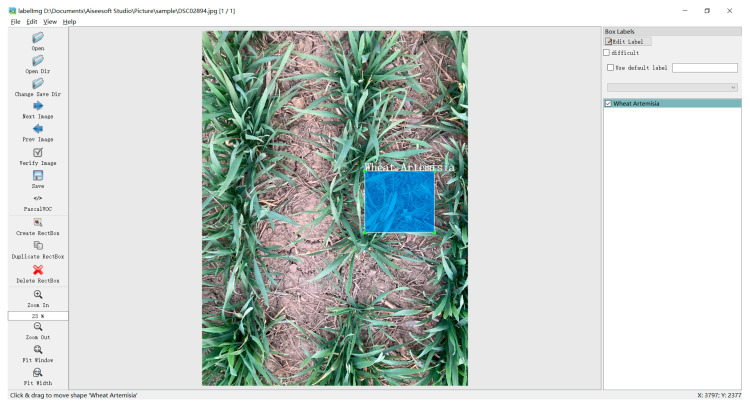
Example of weed labelling (bounding box).

**Figure 3 sensors-24-04379-f003:**
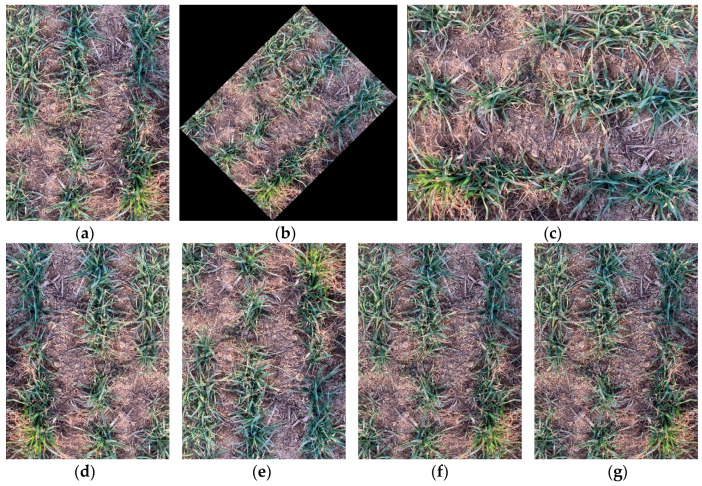
Image data expansion. (**a**) Original images; (**b**) 45° rotation; (**c**) 90° rotation; (**d**) horizontal mirroring; (**e**) vertical mirroring; (**f**) Gaussian noise; (**g**) impulse noise.

**Figure 4 sensors-24-04379-f004:**
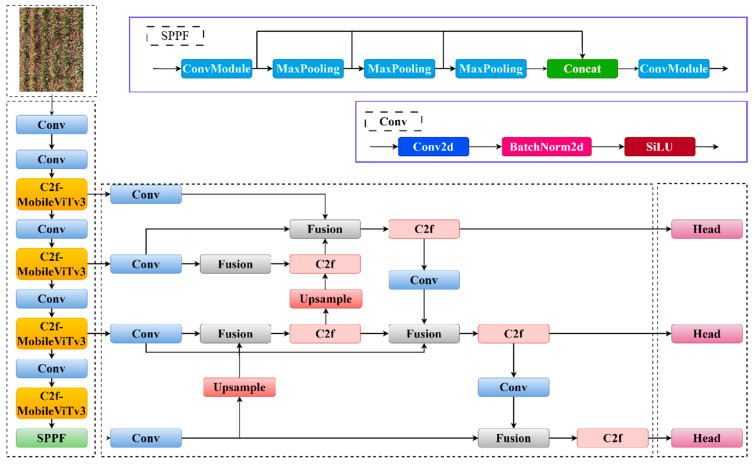
YOLOv8-MBM network architecture.

**Figure 5 sensors-24-04379-f005:**
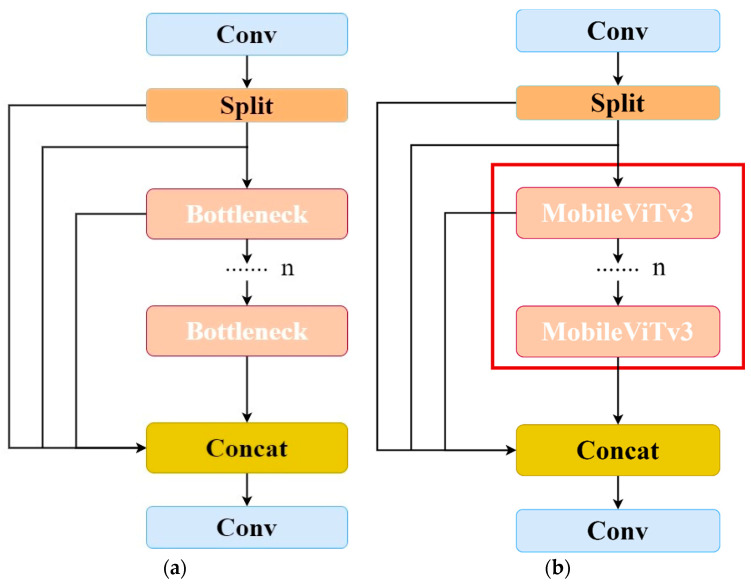
C2f and C2f-MobileViTv3 module structure. (**a**) C2f module structure (**b**). C2f-MobileViTv3 module structure.

**Figure 6 sensors-24-04379-f006:**
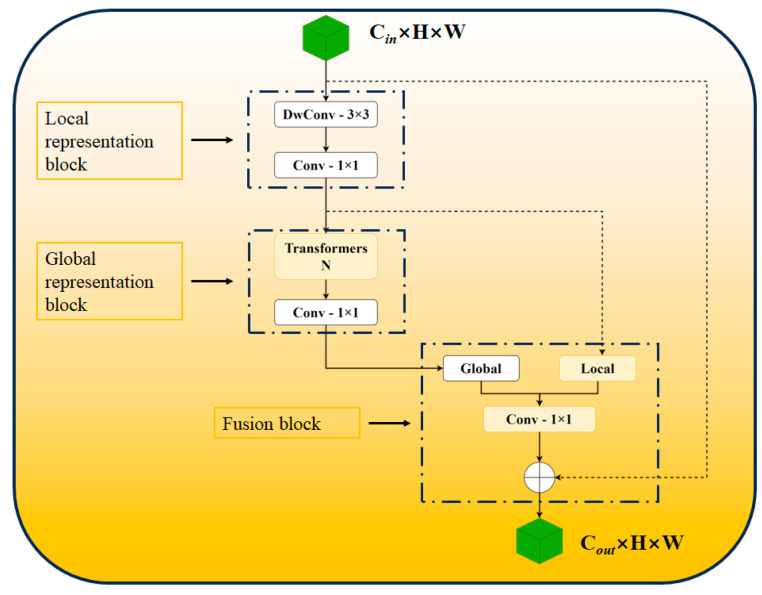
MobileViTv3 structure diagram.

**Figure 7 sensors-24-04379-f007:**
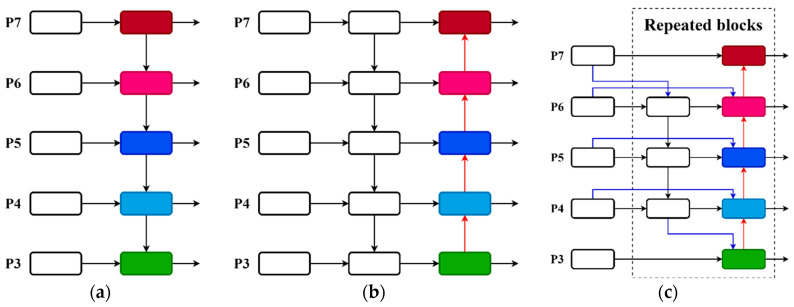
Structure of FPN, PANet, and BiFPN. (**a**) FPN (**b**) PANet (**c**) BiFPN.

**Figure 8 sensors-24-04379-f008:**
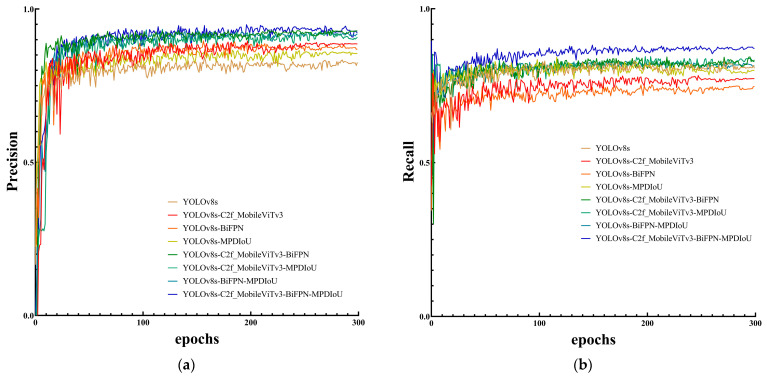
Performance index curves for different combinations of optimisation algorithms.

**Figure 9 sensors-24-04379-f009:**
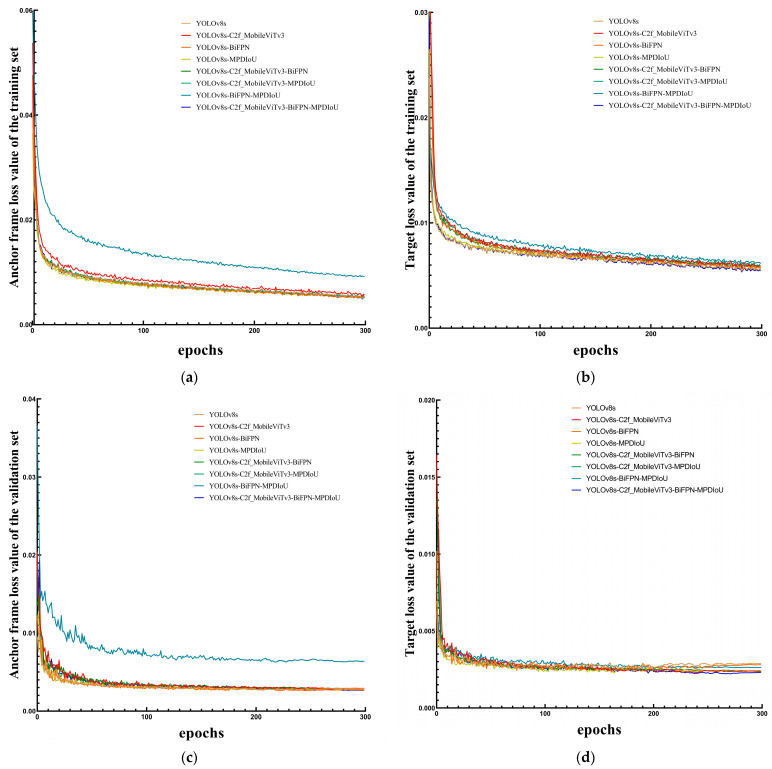
Loss curves for different optimisation algorithms.

**Figure 10 sensors-24-04379-f010:**
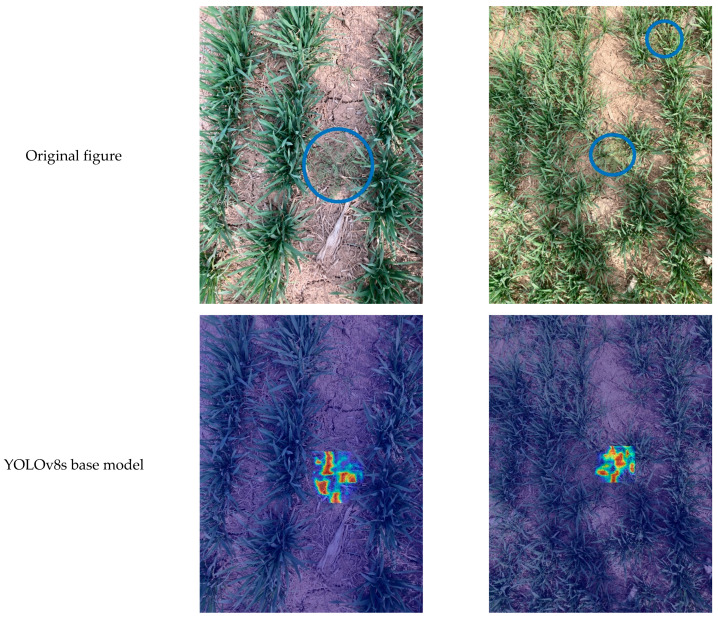
Comparison of heat maps of target areas. Note: Row 1 shows the original collection image, where the blue circle marks the weed.

**Figure 11 sensors-24-04379-f011:**
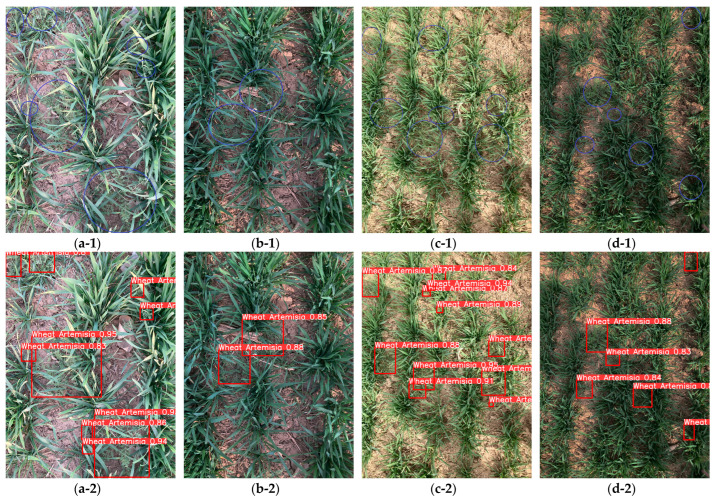
Effectiveness of YOLOv8-MBM algorithm for weed detection. (**a-1**) Original figure with good near-distance light; (**a-2**) detection effect of good near-distance light; **(b-1**) original figure with bad near-distance light; (**b-2**) detection effect of bad near-distance light; (**c-1**) original figure with good long-distance light; (**c-2**) detection effect of good long-distance light; (**d-1**) original figure with bad long-distance light; (**d-2**) detection effect of bad long-distance light. Note: Row 1 shows the original image, where the blue circles mark the weed areas.

**Table 1 sensors-24-04379-t001:** Comparison of the performance of different versions of YOLOv8.

Model	Size	mAP_50_	mAP_50–95_	Parameters/×10 M^6^	Weight Size/MB	FPS
YOLOv8n	640	0.694	0.653	3.2	5.9	42
YOLOv8s	640	0.707	0.759	11.7	21.4	37.9
YOLOv8m	640	0.712	0.734	26.0	49.5	32.3
YOLOv8l	640	0.708	0.721	43.7	83.5	26.1
YOLOv8x	640	0.703	0.719	68.2	130	20.9

**Table 2 sensors-24-04379-t002:** Operating environment.

Configuration	Parameters
CPU	Intel Core i5-12600KF 3.70 GHz
GPUs	NVIDIA GeForce RTX 4060Ti
Operating system	Windows 10
Accelerated environment	CUDA11.6 CUDNN8.9.7
Development environment (computer)	Pycharm 2020.2.1
random access memory (RAM)	32G

**Table 3 sensors-24-04379-t003:** Comparison of ablation test results.

Model	C2f-MobileViTv3	BiFPN	MPDIoU	P/%	R/%	mAP1/%	mAP2/%	Weight Size/MB
YOLOv8s	×	×	×	82.1	80.8	80.0	75.9	21.4
	**√**	×	×	88.7	76.4	84.9	82.3	22.8
	×	**√**	×	87.5	74.9	82.6	78.9	22.6
	×	×	**√**	85.6	80.0	83.8	80.0	22.5
	**√**	**√**	×	92.6	82.8	83.7	82.8	23.0
	**√**	×	**√**	91.1	82.9	88.7	81.5	22.8
	×	**√**	**√**	91.0	82.8	89.8	82.5	22.6
	**√**	**√**	**√**	92.7	89.7	89.7	85.2	23.0

**Table 4 sensors-24-04379-t004:** Comparison test results using mainstream models.

Model	P/%	R/%	mAP1/%	mAP2/%	Weight Size/MB	FPS
YOLOv3	78.0	71.0	73.1	69.4	219.9	34.3
YOLOv4-tiny	78.4	74.6	75.8	75.1	27.6	30.8
YOLOv5s	79.0	71.2	74.0	70.7	14.1	36.7
YOLOv7	81.9	73.2	78.3	72.1	71.2	20.5
YOLOv9	79.6	70.4	78.4	74.3	116.0	11.6
YOLOv8-C2f_M3-BiFPN-MPDIoU	92.7	87.6	89.7	85.2	23.0	35.5

## Data Availability

The datasets generated and analysed during the current study are provided in the article.

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
