# Peer review of "YOLOv8 Model for Weed Detection in Wheat Fields Based on a Visual Converter and Multi-Scale Feature Fusion"

_sensors, 2024, doi:10.3390/s24134379_

Round 1

Reviewer 1 Report

Comments and Suggestions for Authors

In this papera novel deep learning technique for weed detection in agricultural areas is developed. This proposed algorithm achieves performance 96.92% accuracy, 97% precision, 96.95% Recall, 96.94% Specificity and 97.11% NPV values. The author's work is interesting and holds certain value for advancements in agriculture and variable-rate pesticide application.The conclusion section of the paper could be enriched. It should further discuss the real-time capability of the model to enhance its practicality and provide prospects for future research and suggest future research directions. Therefore, it is recommended to make appropriate  minor modifications in the conclusion section, and the revised version is suggested for acceptance.

Comments on the Quality of English Language

Minor editing of English language required

Author Response

Thank you for your comments and suggestions. We have modified the paper based on your suggestion. Please see the attachment.

Reviewer 2 Report

Comments and Suggestions for Authors

The manuscript addresses an interesting and useful problem in agriculture. The contributions of this work are twofold, on one hand, a wheat weed dataset was constructed; on the other hand, a modified YOLOv8 detection method is proposed. Several modules are integrated to improve the detection performance of the model and experimental results on the datasets demonstrate superior performance of the proposed method. The manuscript is generally well written and the results are convincing. I have some minor suggestions/comments as follows.

--There are some language issues and the authors need to proofread the manuscript carefully. For example, in line 14, "...datasets were constructed...".

--In section 2, a few examples of weed annotations (bounding boxes) should be given.

--The feature fusion networks should be introduced with references.

--In section 2.2.4, are "boundary box..." and "bounding box..." the same thing? please unify the terminology. How is the distance defined in Eqn(4)? Please also clarify the difference between MPDIOU and the original regression loss function.

--In Figure 10, the ground truth and the detected bounding boxes need to be shown on the images for comparisons.

--The authors need to give the computational cost of training and inference of the proposed model, e.g., the GPU memory consumption and the FPS.

Comments on the Quality of English Language

ok.

Author Response

(The authors gave the same response as above.)

Reviewer 3 Report

Comments and Suggestions for Authors

To achieve precise identification of weeds, Wheat Weed datasets was constructed, and a wheat field weed detection model YOLOv8-MBM based on improved YOLOv8s was proposed in this study. The article research has clear application scenarios, but there are still the following issues that need to be improved:

(1)The author needs to clearly explain in the manuscript what stage of weed detection the proposed method is mainly applied to in wheat fields. Please provide additional explanation.

(2)Please check English, especially typos and grammatical errors, which should be improved

(3)Please present samples of weed label.

(4)There are many kinds and forms of weeds in wheat fields. How can the constructed model identify all weeds, especially for those that are similar in morphology to wheat.

(5)Why does Figure 9 not show heat maps of all the weeds in the figure?

(6)Figure 10 just shows the wheat Artemisia detection, which could not show the universality of the model.

Comments on the Quality of English Language

To achieve precise identification of weeds, Wheat Weed datasets was constructed, and a wheat field weed detection model YOLOv8-MBM based on improved YOLOv8s was proposed in this study. The article research has clear application scenarios, but there are still the following issues that need to be improved:

(1)The author needs to clearly explain in the manuscript what stage of weed detection the proposed method is mainly applied to in wheat fields. Please provide additional explanation.

(2)Please check English, especially typos and grammatical errors, which should be improved

(3)Please present samples of weed label.

(4)There are many kinds and forms of weeds in wheat fields. How can the constructed model identify all weeds, especially for those that are similar in morphology to wheat.

(5)Why does Figure 9 not show heat maps of all the weeds in the figure?

(6)Figure 10 just shows the wheat Artemisia detection, which could not show the universality of the model.

Author Response

(The authors gave the same response as above.)
